# Research on Potential Network Markers and Signaling Pathways in Type 2 Diabetes Based on Conditional Cell-Specific Network

**DOI:** 10.3390/genes13071155

**Published:** 2022-06-26

**Authors:** Yuke Xie, Zhizhong Cui, Nan Wang, Peiluan Li

**Affiliations:** School of Mathematics and Statistics, Henan University of Science and Technology, Luoyang 471023, China; 191410040118@stu.haust.edu.cn (Y.X.); 191410010105@stu.haust.edu.cn (Z.C.); 191410040115@stu.haust.edu.cn (N.W.)

**Keywords:** CCSN, single-cell, hub gene, ‘dark’ gene, signaling pathway

## Abstract

Traditional methods concerning type 2 diabetes (T2D) are limited to grouped cells instead of each single cell, and thus the heterogeneity of single cells is erased. Therefore, it is still challenging to study T2D based on a single-cell and network perspective. In this study, we construct a conditional cell-specific network (CCSN) for each single cell for the GSE86469 dataset which is a single-cell transcriptional set from nondiabetic (ND) and T2D human islet samples, and obtain a conditional network degree matrix (CNDM). Since beta cells are the key cells leading to T2D, we search for hub genes in CCSN of beta cells and find that *ATP6AP2* is essential for regulation and storage of insulin, and the renin-angiotensin system involving *ATP6AP2* is related to most pathological processes leading to diabetic nephropathy. The communication between beta cells and other endocrine cells is performed and three gene pairs with obvious interaction are found. In addition, different expression genes (DEGs) are found based on CNDM and the gene expression matrix (GEM), respectively. Finally, ‘dark’ genes are identified, and enrichment analysis shows that *NFATC2* is involved in the VEGF signaling pathway and indirectly affects the production of Prostacyclin (PGI_2_), which may be a potential biomarker for diabetic nephropathy.

## 1. Introduction

T2D is a metabolic disorder characterized by hyperglycemia caused by one or both of insulin resistance and insufficient insulin production. People with T2D are at high risk for many serious health-threatening complications, including cardiovascular diseases, blindness, kidney failure, limb amputations, premature death, fractures, frailty, depression, and cognitive decline [1]. Diabetes affected 422 million people and directly caused 4.9 million deaths according to the global report on diabetes in 2014. Therefore, exploring the potential pathogenesis of T2D is of great significance for diabetes prevention and treatment. There has been a great deal of literature on T2D, most of which was based on bulk RNA-seq data, and heterogeneity between single cells was often overlooked.

Single-cell RNA sequencing can uncover unexpected subpopulations, rare cellular states, or novel transcriptional machinery [2,3] and may provide an opportunity to gain insight into cell-specific network systems. The rapid growth of single-cell datasets has shed new light on the complex mechanisms behind cellular heterogeneity. Li et al. used single-cell RNA-seq to establish a comprehensive transcriptome database for the cell types that are present in primary human pancreatic islets and identified human-specific expression patterns in alpha and beta cells [4]. Based on continuous pseudo-time spectrum analysis, Bao et al. found beta cell transition among different cell states [5]. Lawlor et al. identified fundamental cell type-specific features of pancreatic islet function [6]. Segerstolpe et al. revealed subpopulations of alpha, beta, and acinar cells [7].

Single-cell RNA sequencing has promoted the study of cellular heterogeneity and functional diversity. However, most of the current single-cell data algorithms focus on the analysis of gene expression levels. In addition, there are few studies on single-cell gene association networks. To fully extract the information for single-cell data, Li et al. transformed ‘unstable’ gene expression data into ‘stable’ gene association data by constructing a gene–gene direct association network at the single-cell level [8].

In this study, according to the clustering results of dataset GSE86469 by Lawlor et al. [6], we search for hub genes in the CCSN network of beta cells and identify genes with high network correlation. The communication between beta cells and other endocrine cells is analyzed to find gene pairs with obvious interaction. In addition, based on CCSN, we uncover some ‘dark’ genes, which are non-differential in gene expression but sensitive to network degree. We discover that calcium-regulated NFAT protein encoded by ‘dark’ gene *NFATC2* binding to DNA-related sites in the nucleus can induce the expression of COX2, and thereby affects the production of PGI_2_ in the VEGF signaling pathway. *NFAT* has been shown to play an important role in the pathogenesis of diabetes and the development of vascular complications. As a consequence, *NFATC2* may be a new potential biomarker for diabetic nephropathy.

This study has the following advantages:We build a cell-specific direct correlation network from the perspective of gene association based on CCSN, which fundamentally reduces large noise and instability.Based on CCSN, hub genes with high network correlation in CCSN and pathways related to T2D are identified, which can provide more reliable biomarkers for T2D prognosis and treatment.By CCSN method, we are able to uncover ‘dark’ genes, which are non-differential in gene expression but sensitive to the degree value. These genes cannot be found by traditional differential analysis methods and may play important roles in network regulation of T2D. GO and KEGG functional enrichment analysis show that some ‘dark’ genes are involved in T2D-related pathways and might be potential biomarkers for T2D and its complications.

## 2. Materials and Methods

### 2.1. Data Pre-Processing

We download the original dataset GSE86469 from the NCBI (https://www.ncbi.nlm.nih.gov (accessed on 1 October 2021)) website, which is the RNA-seq profile of 638 human pancreatic islet/other single cells obtained from 5 non-diabetic and 3 type 2 diabetic cadaveric organ donors. The original dataset contains 26,616 genes and 638 cells. The 638 cells include 380 single-cell samples from healthy people and 258 single-cell samples from patients with T2D. Lawlor et al. carried out Gaussian mixture modeling (GMM) on the dataset [6]. The 638 cells were clustered into alpha, beta, delta, gamma, ductal, acinar, stellate, and some unrecognized cells as ‘other’. Among them, alpha, beta, delta and gamma belong to endocrine cells. Then we convert the original dataset into GEM with 17420 genes through quality control and gene id conversion. Finally, the logarithm log(1 + *x*) is applied to normalize the GEM with 17,420 rows/genes and 638 columns/cells [9,10]. The pancreatic adenocarcinoma (PAAD) data used in the prognostic analysis came from survival data from UCSC Xena (https://xenabrowser.net/ (accessed on 12 February 2022)).

### 2.2. Construction of CCSN

In this study, CCSN is constructed for each endocrine cell based on the cell-specific network (CSN) proposed by Dai et al. [11] and the CCSN proposed by Li et al. [8].

First, for a given cell k, we choose the largest importance genes as the conditional genes. Then the association between genes is calculated using the statistic constructed by Li et al. [8]. If the statistic is greater than the alpha quantile of the distribution, it means that there is association between genes; otherwise there is no association. Further, we integrate the CCSNs of all cells belonging to the same type into a network. Finally, we transform the network into CNDM for ease of computation. The details of CCSN are provided in the Appendix A.

The flowchart based on the dataset GSE86469 is given in Figure 1.

### 2.3. Identification of Hub Genes Based on CCSN

Insufficient insulin secretion or insulin resistance is an important cause of elevated blood sugar levels. Therefore, by network analysis of insulin-secreting of beta cells, the main genes and pathways related to T2D are identified, which can provide a new understanding of the potential molecular mechanism of disease and provide more reliable biomarkers for T2D prognosis and treatment. Based on the clustering results by Lawlor et al. [6], we extract the GEM of 96 beta cells from the dataset and establish the CCSN of beta cells. The top 20 hub genes associated with T2D, according to the maximum clique centrality (MCC) scores from the cytoHubba plugin in Cytoscape, are screened.

### 2.4. Cell–Cell Communication

Cell–cell communication mediated by ligand-receptor complex is essential for coordinating a variety of biological processes such as development, differentiation and inflammation. Efremova et al. developed CellPhoneDB which can predict abundant cellular interactions between two cell types from single cell transcriptome data [12]. For the genes expressed by the cell population, the percentage of cells expressing the gene and the average value of gene expression are calculated. The cluster tags of all cells are randomly arranged for 1000 times (optional values), and the average expression level of receptors in the cluster and the average expression level of ligands in the interaction cluster are determined. If the observed average is in the top 5%, the p value of the interaction is 0.05. According to the number of significant receptor ligand pairs enriched in the two cell types, the highly specific interactions between cell types are sorted to manually screen the biologically related interactions.

### 2.5. ‘Dark’ Genes Revealed by CNDM

Differential expression analysis is performed using the Bioconductor package edgeR. DEGs of endocrine cells are screened based on GEM and CNDM, respectively. Comparisons between T2D and ND single-cell transcriptomes are performed for the same cell types (e.g., T2D Alpha cells vs. ND Alpha cells). The genes with FDR < 5% are identified as DEGs (Appendix A). DEGs with significant difference in the degree value but have no differential expression are regarded as ‘dark’ genes. In brief, ‘dark’ genes show no significant difference (*p* > 0.05; *t*-test) in the expression level between the diseased and non-diseased states, but there is a significant difference (FDR < 0.05; *t*-test) in the network correlation degree. ‘Dark’ genes are enriched in key functional pathways and perform well in prognosis.

## 3. Results

### 3.1. Hub Gene Analysis

The top 20 ranked nodes are selected using the maximum clique centrality (MCC) method in the cytoHubba plugin in Cytoscape (Table 1). These 20 genes are used as hub genes, and their network relationship is shown in Figure 2. The network of the 20 hub genes is shown with red (high ranking) and yellow nodes (low ranking) based on the ranking score.

GO enrichment results are displayed under the threshold value of 0.01. The 20 hub genes are mainly concentrated in macromolecular complex, extracellular exosome, kinetochore, ribonucleoprotein complex, and mainly participated in the nuclear-transcribed mRNA catabolic process, nonsense-mediated decay, translational initiation and other biological processes (Table 2). A macromolecular complex is a stable set of (two or more) interacting protein molecules. Exosome as a novel biomarker reflecting cell behavior in normal and pathological conditions, e.g., diabetes, is the center of academic attention.

KEGG enrichment results are displayed under the threshold value of 0.05, and 20 hub genes are enriched to 8 pathways (Table 3). In cases of pancreatic beta cell hyperplasia associated with insulin resistance, ribosomal biogenesis is increased. The renin angiotensin system is involved in most of the pathological processes that result in diabetic nephropathy. This system has a central role in the pathophysiology of diabetic nephropathy [13]. Kidney tissues can act on *REN* indirectly through the Renin pathway, which induces *ATP6AP2* to promote protein synthesis and anti-apoptosis and regeneration. In addition, *ATP6AP2* is critical for regulating the stored insulin pool and a balanced regulation of granule turnover is key to maintaining beta cell function and diabetes prevention. *ATP6AP2* deficiency is the exacerbated generation and accumulation of multigranular bodies that consume the cytoplasm of beta cells, ensnaring insulin secretory granules (SGs) and thus causing insulin-deficient diabetes [14]. We find that there is Coronavirus disease-COVID-19 in the enriched pathway. SARS-CoV-2 is known to infect human beta cells and possibly alters islet function, suggesting that diabetics are at high risk of contracting COVID-19. Diabetes may be a risk factor for salmonellosis due to decreased gastric acidity and prolonged gastric transit time [15]. *HSP90AB1* and *RPS3* can activate a DNA molecule to *TNFα* directly or indirectly involved in the NF-κB pathway. The IKK/NF-κB pathway plays a critical role in the induction and maintenance of the inflammatory state underlying metabolic diseases such as obesity and T2D [16]. *HSP90AB1* has been identified as a potential therapeutic target for metabolic diseases including diabetes, and suppression of *HSP90AB1* is a valid therapeutic clinically relevant strategy in the management of dysregulated metabolic disease and insulin resistance [17]. The gene pathway association network diagram in functional enrichment analysis is shown in Figure 3.

In addition, we conduct prognostic analysis of hub genes, and the results are shown in the Appendix A.

### 3.2. Cell–Cell Communication Analysis

The communication between beta and the other three endocrine cell types shows that the interaction between *INS_INSR* (insulin receptor), *INS_IDE* (insulin degrading enzyme) and *INS_LILRB1* (leukocyte immunoglobulin-like receptor) is obvious (Figure 4). We discover that these genes interacting significantly with the insulin (*INS*) gene are directly or indirectly involved in the development of T2D.

*INS* is responsible for the production of insulin from beta cells of the pancreas, and the binding of insulin or other ligands to this receptor (*INSR*) activates an insulin signaling pathway that regulates glucose uptake and release. Any functional defect of *INSR* gene will directly affect the action of insulin and cause insulin resistance, thus leading to the development of T2D [18]. *IDE* gene encodes a zinc metallopeptidase that degrades intracellular insulin, and thereby terminates insulin activity, as well as participating in intercellular peptide signaling by degrading diverse peptides such as glucagon, amylin, bradykinin, and kallidin. The preferential affinity of this enzyme for insulin results in insulin-mediated inhibition of the degradation of other peptides such as beta-amyloid. Deficiencies in this protein’s function are associated with Alzheimer’s disease and T2D. *LILRB1* is also called *microRNA7* (*miR-7*). *MiR-7* is an evolutionarily highly conserved miRNA and considered to be a typical neuroendocrine miRNA. *MiR-7* is highly expressed in neuroendocrine organs such as the pancreas and brain, which can regulate important aspects of pancreatic biology and function. Potential *miR-7* targets are enriched in insulin signal transduction, which indicate that the gene may indirectly affect regulation of insulin [19].

### 3.3. ‘Dark’ Gene Analysis

Based on GEM and CNDM, we find 13 ‘dark’ genes (Appendix A). Figure 5 shows the differences of gene expression and degree value of ‘dark’ genes *FAM189A2* and *TNFAIP6* between ND and T2D. The results showed that for the ‘dark’ genes, there are no significant differences at the gene expression level, but significant changes (FDR < 0.05; log-rank text) are observed at the network correlation degree level.

#### 3.3.1. Prognostic Analysis of ‘Dark’ Genes

The prognostic analysis of chronic diseases has great biological significance. However, due to the lack of corresponding prognostic data in the study of T2D at present, the prognostic analysis of T2D cannot be performed directly. It is reported that diabetes is associated with an increased risk of PAAD in both males and females and that diabetes mellitus is both an early manifestation and an etiologic factor of PAAD [20]. To explore the effect of ‘dark’ genes on patients with diabetes, we use the data of PAAD, which is closely related to T2D to analyze the prognosis of T2D, in order to reveal the mechanism of the development of T2D from another viewpoint.

Firstly, we analyze the prognosis of these ‘dark’ genes, respectively, based on gene expression and degree value by dividing the samples into two groups based on the median of genes’ expression or degree value. The high group is a group with higher value and the low group is a group with a lower value. Secondly, based on the result of prognosis, the ‘dark’ genes can be categorized into two types of molecules as a mutual marker for all samples. Those genes with high scores that cause poor prognosis are termed “negative dark genes”, and those genes with high scores that cause good prognosis are termed “positive dark genes”. If “negative dark genes” appear in a sample, the prognosis would be more negative than that of other samples. Similarly, if “positive dark genes” appear in a sample, the prognosis would be more positive.

For PAAD, Figure 6 shows that ‘dark’ genes play an important role in the prognosis of patients with PAAD. For *SLC43A1*, at the level of gene expression, the effect on the survival rate of patients is very small, which does not reach statistical significance (*p* = 0.54 > 0.05; log-rank test), but in the level of degree value of genes, the survival rate of patients is significantly different (*p* = 0.025 < 0.05; log-rank test). It is obvious that the survival rate of patients with high *SLC43A1* degree value is significantly higher than that of patients with low degree value, which is associated with “positive dark genes”. The prognostic results mean that *SLC43A1* has potential application value in accurate medical treatment or personalized treatment of pancreatic problems.

For *SLC43A1*, elevated concentrations of circulating branched-chain amino acids (BCAA) have emerged as an early predictive/prognostic indicator of PAAD development [21]. Furthermore, elevated circulating BCAA levels are also associated with numerous states characterized by an inflammatory response and insulin resistance (IR), namely obesity and diabetes [22,23]. Based on the study, elevated circulating levels of BCAA are associated with insulin resistance and incident T2D, and BCAA may be useful biomarkers for monitoring the early response to therapeutic interventions for T2D [24]. BCAA transport into the cell across the plasma membrane is primarily carried out by a heterodimeric protein known as the large neutral amino acid transporter (LNAA) [25]. LNAA consists of a 55 kDa *SLC7A5* subunit (subunit 1, LAT1, CD98 light chain) and a 68 kDa *SLC3A2* subunit (subunit 2, CD98 heavy chain) [26]. BCAA transporters heterodimerizing with *SLC3A2* include *SLC7A8* (LAT2), *SLC43A1* (LAT3), and *SLC43A2* (LAT4) [27]. Therefore, *SLC43A1* (LAT3), as a part of BCAA transporter, participates in the transport of elevated circulating BCAA and has some connection with T2D.

#### 3.3.2. ‘Dark’ Gene Pathway Analysis

Under the threshold value of 0.05, 13 ‘dark’ genes are mainly enriched in filopodium. In terms of molecular function, this is mainly related to carboxylic ester hydrolase activity (Table 4). Filopodium are observed at the advancing front of the migrating cell and are implicated in cell motility as well as in cell-substrate adhesion. Carboxylic ester hydrolase is a serum marker of acute pancreatitis, and recurrent pancreatitis can lead to diabetes. Carboxylic ester hydrolases act on ester bonds (EC 3.1.1.) and are often applied in several biotechnological processes. In addition, ‘dark’ genes enriched six KEGG pathways, including the Axon guidance, VEGF signaling pathway, etc. (Table 5). Islet-expressed axon-guiding molecules possess essential cell–cell connectivity for the maintenance of normal islet function in adulthood. Vascular endothelial growth factor (VEGF) is one of the major factors promoting diabetic retinopathy (DR) [28]. Previous studies have demonstrated impaired phagocytosis of FC-γ receptors in monocytes of T2D patients with chronic hyperglycemia [29].

### 3.4. The Underlying Signaling Mechanisms Revealed by ‘Dark’ Genes NFATC2 and UNC5D

To further clarify the association of ‘dark’ genes and DEGs in the pathway, we focus KEGG analysis on the two pathways most associated with the progression of T2D and its complications, namely Axon guidance and VEGF signaling pathway. The circulating axon guidance pathway (AGP) proteins are associated with risk of end stage kidney disease (ESKD), and diabetic nephropathy as the leading cause of ESKD is one of the most common long-term microvascular complications of diabetes mellitus. There are several signaling pathways that are stimulated in diabetes and potentially cross-talk to help each other control *VEGF* release. *VEGF-A* is associated with the development of diabetic nephropathy.

#### 3.4.1. ‘Dark’ Gene *NFATC2* Is a Key Transcription Factor of Axon Guidance and VEGF Signaling Pathway Related to Diabetes Complications

We discover that ‘dark’ gene *NFATC2* is involved in Axon guidance and VEGF signaling pathway and is a common key transcription factor in these two pathways. The *NFAT* family of transcription factors is composed of five members, *Nfatc1-4* and *Nfat5*, expressed in pancreatic islets where they are thought to integrate calcium signals to coordinate gene expression and regulate growth, differentiation and cellular response to environmental cues [30]. *NFAT* has been shown to play an important role in the pathogenesis of diabetes and the development of vascular complications. *NFATC2* is a key regulator of beta cell proliferation and function. Simonett et al. identified approximately 250 direct transcriptional targets of *NFAT* in human islets [31].

In Axon guidance pathway, *Netrin-1* binds with *DCC* multimer to further activate PLCγ, indirectly acting on Ca^2+^ to activate CaN to dephosphorylate *NFAT* (*NFATC2*) (Figure 7a and Figure 8). Netrin is a member of the laminin-like protein family and may be extensively involved in the regulation of angiogenesis, inflammation, tissue remodeling, and cancer. *DCC* is established as a receptor for *netrin-1*, and PLCγ1 plays a role in axon extension and guidance by mediating netrin-1/DCC signaling. CaN (PPP3C) is the catalytic subunit of Protein Phosphatase 2B (PP2B) holoenzyme (aka calcineurin), and activates a vertebrate-specific transcription factor called NFATc. Calcineurin/NFAT signaling pathway is important in axonal growth and guidance during vertebrate development. Diabetes has been shown to cause insufficient axonal growth [32]. The extension and organization of sensory axon projection and commissural axon growth are both dependent upon *NFAT* activity.

In VEGF signaling pathway, *VEGF*(*VEGFA*) activates the receptor VEGFR2(*KDR*) and binds to it, PLCγ1 as a signal transducer converts an extracellular stimulus into intracellular signals by generating inositol-1,4,5-trisphosphate (IP_3_), and IP_3_ and Ca^2+^ induce CALN to dephosphorylate *NFAT* (*NFATC2*). NFAT proteins and their activation in the nucleus and binding to DNA-related sites can easily induce the expression of downstream target genes, PGI_2_ is metabolized from arachidonic acid via cyclooxygenase (COX)-1 and COX-2 in an initial metabolic step (Figure 7b and Figure 8). *VEGF* family members are crucial to the normal development and maintenance of the vascular and lymphatic systems. In the kidney, *VEGF* is almost exclusively expressed in glomerular and tubular epithelial cells, while the VEGF type-2 receptor (*VEGF-R2*)/*KDR* is mainly present in glomerular and tubular endothelial cells, but also in interstitial cells. Upregulation of *VEGFA* in diabetic kidneys protects the microvasculature from injury and that reduction of *VEGFA* in diabetes may be harmful [33]. *VEGFs* and its receptor *VEGFR-2* (*KDR*) have a significant impact on the process of angiogenesis. Disturbances in physiologic angiogenesis can cause diabetic retinopathy and nephropathy and inhibit angiogenesis in transplant rejection in diabetic recipients. In T2D, glomerular lesions are created in the form of an increase in glomerular endothelial cell number in consequence of imbalance in cell proliferation and apoptosis. The main role in this process is attributed to *VEGF-A* expression following high glucose levels in the early phases of diabetes [34]. *VEGFR-2* inhibitors prevent angiogenesis and lymph angiogenesis, and some biologics such as ramucirumab (Cyramza), bevacizumab (Avastin), ranibizumab (Lucentis) have been approved to target VEGF and VEGF receptors [35]. Evidence suggests that diabetes mellitus is associated with PGI_2_ dysregulation; reduced PGI_2_ production by the vascular wall has been proposed as a possible cause of macro- or microangiopathy in diabetes mellitus [36]. Prostacyclin synthase (PGIS) deficiency induced renal fibrosis along with the notable irregulation of renal hemodynamics, tubular atrophy, surface irregularities and cysts, and overexpression of PGIS contributed to the renal protection against endotoxemia-related Acute kidney injury (AKI). Studies in subcutaneous arteries from patients with diabetes mellitus suggest that COX-2 expression is increased, whereas prostacyclin synthase expression is decreased [37]. Although selective inhibition of COX-2 can improve the endothelial dysfunction, it has the potential harm of a decreased PGI_2_ level. Prostacyclin is downregulated in patients with diabetes mellitus who are at high risk of cardiovascular disease, kidney disease and other diseases. Once-daily treatment and standby PGI_2_ may provide the greatest degree of protection. *NFATC2*, which can indirectly affect the output of PGI_2_ and related signaling pathways, provides new therapeutic targets for diabetic nephropathy. *NFATC2* may be a new potential biomarker of diabetic nephropathy.

#### 3.4.2. ‘Dark’ Gene *UNC5D* Is Involved in a Potential Signaling Pathway in the Development of Diabetes-Related Complications Axon Guidance

*Netrin-1* forms a ternary complex with *DCC* and *UNC-5* complexes, the ternary complex binding to *Shp2* to activate the signal transduction pathway for repulsion (Figure 7a). *Netrin-1* seems to cluster different receptors together, leading to alternative signaling outcomes. One such receptor is *UNC5*. Therefore, *DCC* plays a key role in a molecular switch to turn attraction to repulsion. *UNC5* dependent repulsion requires the presence of *DCC* to be expressed on the individual growth cone [38]. *UNC-5* receptors have been proposed as putative tumor suppressor genes. More and more attention has been paid to the inhibition of *UNC5* receptor and its biological function in human malignant tumors, *UNC5D* could be a potential diagnostic biomarker and therapeutic target for metastatic prostate cancer (PCa) [39]. However, the mechanism of *UNC5D* expression in diabetes is still unclear.

## 4. Discussion

In this study, inspired by the CCSN method [8] and the clustering results of dataset GSE86469 [6], we construct a CCSN for each single cell and search for hub genes in the CCSN network of beta cells to find genes with high network correlation. We find that *ATP6AP2* is critical for the regulation of insulin and is key to the maintenance of beta cell function and the prevention of diabetes. *HSP90AB1* may serve as a potential new target for the treatment of metabolic diseases, including diabetes. The analysis of cell–cell communication between beta cells and other endocrine cells shows that there were three gene pairs with obvious interaction, namely *INS_INSR*, *INS_IDE* and *INS_LILRB1*. In addition, CNDM and GEM are used to find out ‘dark’ genes. Prognostic analysis and enrichment analysis of ‘dark’ genes show that *NFATC2* indirectly affects PGI_2_ production in VEGF signaling pathway, which has a certain impact on the treatment of diabetic nephropathy and can be used as a potential therapeutic drug target.

The construction of CCSN can more reliably describe cell types from the perspective of gene association, identify hub genes and explore their roles in network regulation. Cell–cell communication mediated by ligand receptor complex intercellular communication for coordination of various biological processes is crucial. In the biomedical field, DEGs are important for the discovery of new biomarkers, regulators and drug targets. However, some non-DEGs may also be involved in important biological processes and should not be ignored. By CNDM, we can reveal these ‘dark’ genes, and some ‘dark’ genes are enriched in key functional pathways and perform well in prognosis [40]. Due to the lack of corresponding prognostic data in the study of T2D, we use the data of PAAD, which is closely related to T2D to analyze the prognosis of T2D, in order to reveal the mechanism of the development of T2D from another viewpoint. But the outcome will not be able to directly reflect the ‘dark’ genes’ effect T2D.

## 5. Conclusions

In the present study, by constructing CCSN for each single cell for the GSE86469 dataset which is a single-cell transcriptional set from ND and T2D human islet samples, we can have a full understanding of the pathogenesis of T2D from the single-cell level and network perspective. Research on T2D-related pathways involved in ‘dark’ genes can help us explore potential signaling pathways and therapeutic potential biomarkers, which may be helpful for the treatment and prevention of T2D.

## Figures and Tables

**Figure 1 genes-13-01155-f001:**
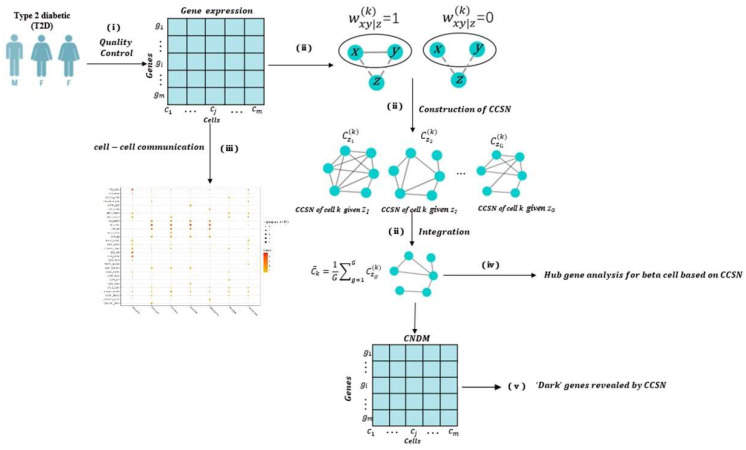
The flow chart of this study. (**i**) We convert the original dataset into GEM with 17,420 genes through quality control and gene id conversion. (**ii**) CCSN is constructed from GEM of different cell types; (**iii**) Cell–cell communication is carried out for endocrine cells; (**iv**) The analysis of hub genes. (**v**) The analysis of ‘dark’ genes.

**Figure 2 genes-13-01155-f002:**
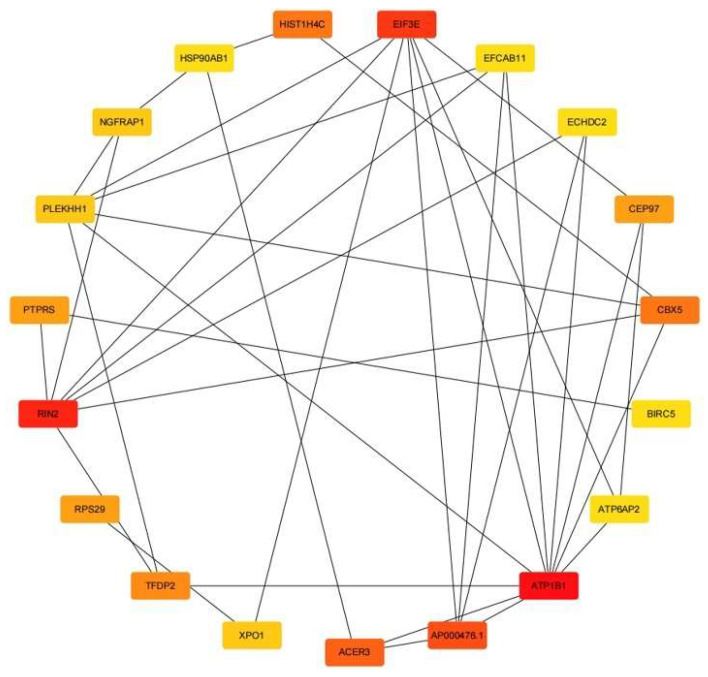
The network of the 20 hub genes is shown with red (high ranking) and yellow nodes (low ranking) based on the ranking score.

**Figure 3 genes-13-01155-f003:**
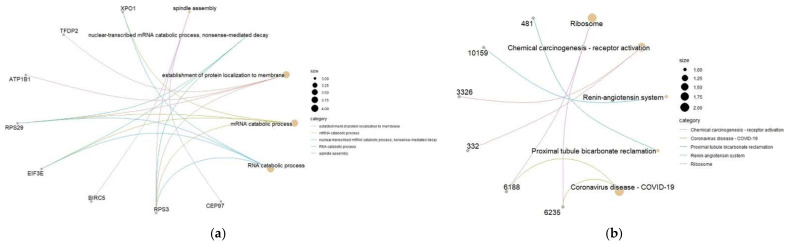
(**a**) GO gene-pathway association network diagram. (**b**) KEGG gene-pathway association network diagram; the numbers represent ENTREZID.

**Figure 4 genes-13-01155-f004:**
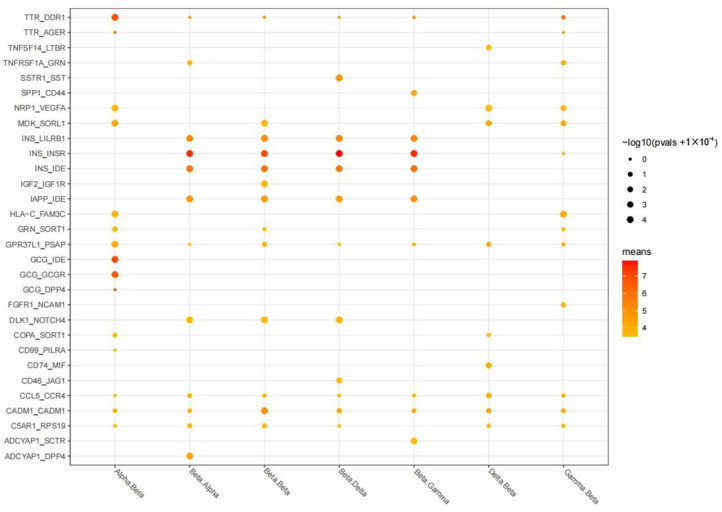
Bubble diagram of the interaction between beta cells and the other three endocrine cell types. Each column is two cell types (such as Alpha.Beta), and each row is the name of the receptor ligand (such as *INS_INSR*). The red color represents the average expression level of the two genes within the two cells. The redder color represents the higher expression value, the size of the bubble represents the −log10 value of the *p* value, and the larger the bubble, the more significant the interaction.

**Figure 5 genes-13-01155-f005:**
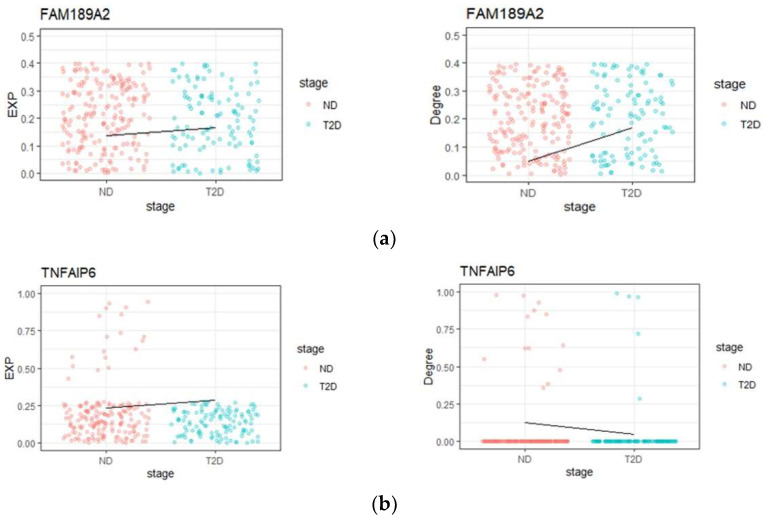
The ‘dark’ genes sensitive to conditional network correlation degree. (**a**) *FAM189A2*; (**b**) *TNFAIP6*. The degree values (right) and gene expression levels (left) of ‘dark’ genes for ND-T2D are provided.

**Figure 6 genes-13-01155-f006:**
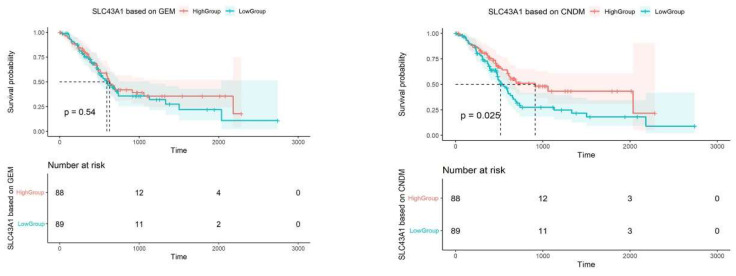
The prognosis curve of ‘dark’ gene *SLC43A1*. According to Kaplan-Meier mapping, the prognostic significance of ‘dark’ genes in patients with pancreatic cancer was shown. High group represents patients with high expression, and Low group represents patients with low expression.

**Figure 7 genes-13-01155-f007:**
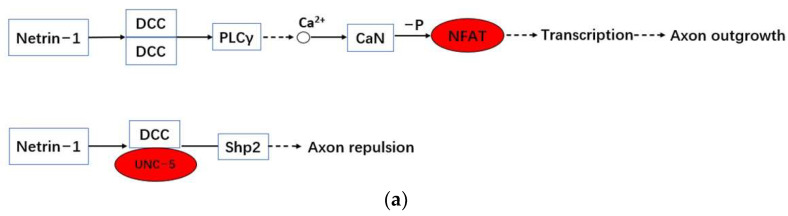
The potential signaling mechanisms of ‘dark’ genes. (**a**) The enrichment and regulation of related ‘dark’ genes, DEG of GEM and CNDM in Axon Guidance; (**b**) The enrichment and regulation of related ‘dark’ genes, DEG of GEM and CNDM in VEGF signaling pathway.

**Figure 8 genes-13-01155-f008:**
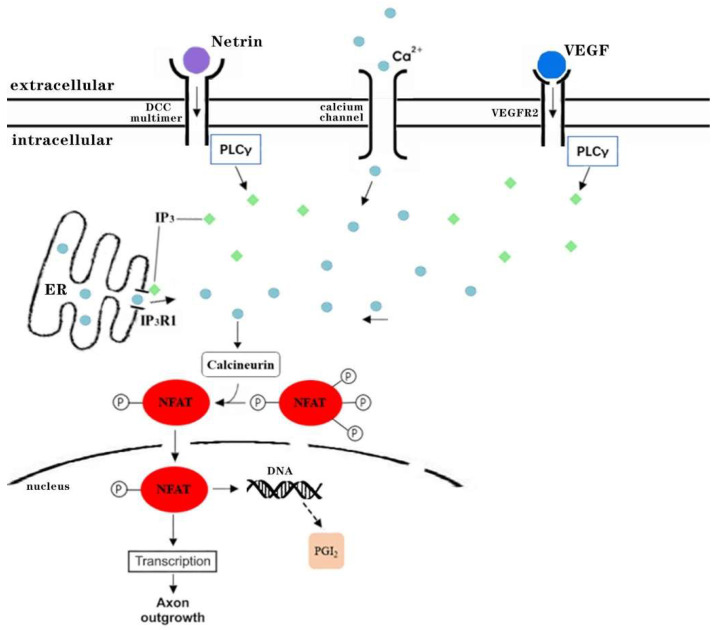
*NFATC2* indirectly affects major biological processes in Axon outgrowth and PGI_2_ production in related signaling pathways.

**Table 1 genes-13-01155-t001:** The top 20 ranked nodes are selected using the MCC method in the cytoHubba plugin in Cytoscape.

Rank	Name	Score
1	*ATP1B1*	39
2	*RIN2*	22
3	*EIF3E*	18
4	*AP000476.1*	15
5	*ACER3*	11
6	*CBX5*	10
6	*HIST1H4C*	10
8	*TFDP2*	9
9	*RPS29*	8
9	*PTPRS*	8
9	*CEP97*	8
12	*RPS3*	7
13	*NGFRAP1*	6
13	*XPO1*	6
13	*PLEKHH1*	6
16	*ECHDC2*	5
16	*EFCAB11*	5
16	*BIRC5*	5
16	*ATP6AP2*	5
16	*HSP90AB1*	5

**Table 2 genes-13-01155-t002:** GO Analysis of Hub Genes.

Enriched Biological Process	Enriched *p* Value
macromolecular complex (GO:0032991)	0.001387
nuclear-transcribed mRNA catabolic process, nonsense-mediated decay (GO:0000184)	0.003306
extracellular exosome (GO:0070062)	0.003644
translational initiation (GO:0006413)	0.003925
Kinetochore (GO:0000776)	0.004606
ribonucleoprotein complex (GO:1990904)	0.006289

**Table 3 genes-13-01155-t003:** KEGG Analysis of Hub Genes.

Enriched Biological Process	Enriched *p* Value
hsa03010: Ribosome	0.015199
hsa05207: Chemical carcinogenesis—receptor activation	0.026454
hsa04614: Renin-angiotensin system	0.027904
hsa04964: Proximal tubule bicarbonate reclamation	0.027904
hsa05171: Coronavirus disease—COVID-19	0.031280
hsa05132: Salmonella infection	0.035643

**Table 4 genes-13-01155-t004:** GO Analysis of ‘Dark’ Genes.

Enriched Biological Process	Enriched *p* Value
carboxylic ester hydrolase activity (GO:0052689)	0.018599
filopodium (GO:0030175)	0.046597

**Table 5 genes-13-01155-t005:** KEGG Analysis of ‘Dark’ Genes.

Enriched Biological Process	Enriched *p* Value
hsa04360: Axon guidance	0.002895
hsa04370: VEGF signaling pathway	0.028677
hsa04662: B cell receptor signaling pathway	0.039688
hsa05235: PD-L1 expression and PD-1 checkpoint pathway in cancer	0.043020
hsa04658: Th1 and Th2 cell differentiation	0.044445
hsa04666: Fc gamma R-mediated phagocytosis	0.046818

## Data Availability

Dataset GSE86469 from the NCBI (https://www.ncbi.nlm.nih.gov (accessed on 1 October 2021)). The PAAD data used in the prognostic analysis came from survival data from UCSC Xena (https://xenabrowser.net/ (accessed on 12 February 2022)). The original code are available at https://github.com/xykxingchen/T2D-main.git (accessed on 17 May 2022).

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
