# Peer review of "Research on Potential Network Markers and Signaling Pathways in Type 2 Diabetes Based on Conditional Cell-Specific Network"

_genes, 2022, doi:10.3390/genes13071155_

Round 1
Reviewer 1 Report
This is an interesting research topic in cell network. The statistical results may be helpful for the treatment and prevention of T2D. The three advantages provide their contribution in the field.
I am not expert in the gene identification. My main concern is the English writting issue. Too many past tense are used in the paper, including the work has done in the literature or the authors' work. It's difficult to see their own contributions in the field.
In page 2, line 49, "To fully mine the information" is not normal, or it may spell wrong?
In Sec. 2.2, Construction of CCSN. Better to show the diffierence in yours and the one in Ref[8] since the work is based on Ref8].
In Page 3, Line 1, "greater" should be is greater or is higher?
"are in" —> are provided in
Author Response
Point 1: Too many past tense are used in the paper, including the work has done in the literature or the authors' work. It's difficult to see their own contributions in the field.
Response 1: We are sorry that our description has puzzled you. In light of your comment, we changed the our work in this paper to present tense.
Point 2: In page 2, line 49, "To fully mine the information" is not normal, or it may spell wrong?
Response 2: In light of your comment, we changed "To fully mine the information" to “To fully extract the information”.
Point 3: In Sec. 2.2, Construction of CCSN. Better to show the difference in yours and the one in Ref[8] since the work is based on Ref8].
Response 3: Compared with Ref[8], except for the different research objects, we integrate the CCSNs of all cells belonging to the same type into a network. In Ref[8], Li et al. constructed a CCSN for each neural progenitor cell and performed network analysis on a single neural group cell. In our paper, we first construct 96 CCSNs for 96 islet beta cells respectively. Then we further integrate these 96 CCSNs into a network, so we analyze the network of islet beta cells of the same type as a whole.
Point 4: In Page 3, Line 1, "greater" should be is greater or is higher?
"are in" —> are provided in
Response 4: Thank you for you reminding. Through careful review of the manuscript, the English expression you mentioned has been corrected in the manuscript.
Thank you again for your positive and constructive comments and suggestions on our manuscript.
Reviewer 2 Report
I Consider that authors could be increase the resolution of figure 6 for better presentation of the paper.
Author Response
Point 1: I Consider that authors could be increase the resolution of figure 6 for better presentation of the paper.
Response 1: Thank you for you reminding. We improved the resolution and sharpness of figure 6.
Thank you again for your positive and constructive comments and suggestions on our manuscript.
Reviewer 3 Report
This is a very interesting and important piece of science. The focus on cell-cell interaction and the 'dark' genes is of utmost importance to overcome two basic limitations of standard omics approach: namelly the overlloking of the basic biological facts of tissues as integrated dyanmical systems whose behaviour emerges from the interaction of its constituents and the presence of gene regulation networks (GRNs) that involve genes not necessarily having a different level of expression in diverse conditions but crucial in the wiring structure of GRNs.
The authors suceed in this task by a thorough analysis of different cell types gene correlation networks. I have only a suggestion for the authors: instead of using the number of links as 'hub signature' why not try and explore betweeness or other centality measures (closeness, eigenvector centrality) to pick up genes most involved in signaling ?
Again compliments for a very novel and inspiring work.
Author Response
Point 1: instead of using the number of links as 'hub signature' why not try and explore betweeness or other centality measures (closeness, eigenvector centrality) to pick up genes most involved in signaling ?
Response 1: In this paper, a topology analysis method (Maximal Clique Centrality) in cytoHubba is used to find the hub gene, rather than the number of edges of the node. cytoHubba is known to rank nodes according to their properties in the network, and it provides 11 topological analysis methods, including several methods you mentioned. In the article [1], the comparison results of these methods shown that Maximal Clique Centrality method is the best. Therefore, the hub gene we obtained using this method is very convincing.
Thank you again for your positive and constructive comments and suggestions on our manuscript.
Reference
[1] Chin, C.H.; Chen, S.H.; Wu, H.H.; Ho, C.W.; Ko, M.T.; Lin, C.Y. cytoHubba: identifying hub objects and sub-networks from complex interactome. BMC Syst Biol. 2014, 8, Suppl 4(Suppl 4):S11.
Round 2
Reviewer 1 Report
The modified version is acceptable now